# Myeloperoxidase as a Marker to Differentiate Mouse Monocyte/Macrophage Subsets

**DOI:** 10.3390/ijms23158246

**Published:** 2022-07-26

**Authors:** Cody J. Gurski, Bonnie N. Dittel

**Affiliations:** 1Versiti Blood Research Institute, Milwaukee, WI 53226, USA; cgurski@versiti.org; 2Department of Microbiology and Immunology, Medical College of Wisconsin, Milwaukee, WI 53226, USA

**Keywords:** macrophage, monocyte, myeloperoxidase

## Abstract

Macrophages are present in every tissue in the body and play essential roles in homeostasis and host defense against microorganisms. Some tissue macrophages derive from the yolk sac/fetal liver that populate tissues for life. Other tissue macrophages derive from monocytes that differentiate in the bone marrow and circulate through tissues via the blood and lymphatics. Circulating monocytes are very plastic and differentiate into macrophages with specialized functions upon entering tissues. Specialized monocyte/macrophage subsets have been difficult to differentiate based on cell surface markers. Here, using a combination of “pan” monocyte/macrophage markers and flow cytometry, we asked whether myeloperoxidase (MPO) could be used as a marker of pro-inflammatory monocyte/macrophage subsets. MPO is of interest because of its potent microbicidal activity. In wild-type SPF housed mice, we found that MPO^+^ monocytes/macrophages were present in peripheral blood, spleen, small and large intestines, and mesenteric lymph nodes, but not the central nervous system. Only monocytes/macrophages that expressed cell surface F4/80 and/or Ly6C co-expressed MPO with the highest expression in F4/80^Hi^Ly6C^Hi^ subsets regardless of tissue. These cumulative data indicate that MPO expression can be used as an additional marker to differentiate between monocyte/macrophage subsets with pro-inflammatory and microbicidal activity in a variety of tissues.

## 1. Introduction

Myeloperoxidase (MPO) is a heme peroxidase that is known to be expressed at high levels in neutrophils [1]. Upon activation, neutrophils generate superoxide anions, which may then generate other reactive oxygen species (ROS), such has hydrogen peroxide (H_2_O_2_) [1,2]. MPO catalyzes the formation of hypochlorous acid (HOCl) from H_2_O_2_ and chloride ions [3], which functions as a potent microbicidal compound, making MPO an important component of the innate immune system [4]. The bactericidal activity of HOCl is a consequence of its ability to alter amino acids, lipids, and DNA [5,6]. Such modifications result in the irreversible oxidation and chlorination of amino acids, the formation of phospholipid chlorohydrins, and nucleotide chlorination, which subsequently inhibits or unfolds proteins, disrupts cell membrane structure, and dissociates double-stranded DNA, respectively [7]. In addition, through the formation of neutrophil extracellular traps (NETs), neutrophils release chromatin that is bound with granule proteins, including MPO, in a web-like structure that is capable of binding and killing microbes independent of phagocytosis [8,9]. NETs are especially useful in the case of fungi, which are difficult to phagocytize [6,10,11]. MPO-deficient individuals cannot form NETs, leaving them more vulnerable to fungal infections such as *Candida albicans* [6,7,8].

MPO has also been described in monocytes [12,13] and in some macrophage populations [13,14], although these populations have not been thoroughly defined. Collectively, macrophages are a diverse population that reside in every tissue, thereby eliciting tissue-specific homeostasis and immune functions. The ubiquitous nature of macrophages also makes them likely to play roles in most disease processes [15]. Macrophage phagocytosis is important for pathogen clearance and, if coupled with MPO expression, would provide superior microbicidal protection. However, as with neutrophils, macrophage release of MPO could drive tissue damage. MPO and the reactive species that it produces have been implicated as pathogenic to surrounding tissues in cases of multiple sclerosis, Alzheimer’s disease, arthritis, inflammatory bowel disease, and other inflammatory diseases [16,17,18,19,20,21,22].

Macrophage subsets lack explicit markers, making them difficult to phenotype and identify. In addition, macrophages are very plastic, allowing them to respond to environmental cues during homeostasis and disease. Markers such as F4/80, Ly6C, CD64, CD169, CD209, MerTK, and more, have often been described as markers of various macrophage subsets [23,24,25,26,27,28,29,30,31]. However, many of these markers have also been shown to be expressed by dendritic cells, eosinophils, and other myeloid cell types, which makes macrophage categorization challenging. Thus, additional markers are needed to differentiate between specific subsets.

Here, we phenotyped macrophage subsets by flow cytometry in peripheral blood, spleen, central nervous system (CNS), large and small intestine, and mesenteric lymph nodes (MLN) to discern whether MPO could be used to differentiate macrophage subsets in the steady state. We found that in all tissues that were studied, except for the CNS, monocytes/macrophages generally expressed intermediate to low levels of MPO. Only the small intestine contained a macrophage subset expressing very high levels of MPO. Collectively these data demonstrate that MPO is an additional marker of monocyte/macrophage subsets that likely exhibit potent anti-microbial function.

## 2. Results

### 2.1. Eosinophils Exhibit MPO False Positive Staining

We utilized flow cytometry to determine whether MPO expression could be used to differentiate monocyte/macrophage populations in the blood, spleen, central nervous system (CNS), large and small intestine, and mesenteric lymph nodes (MLN). For this study, the term monocyte is used for cells that are circulating in the blood and lymphatics and macrophage for the spleen, CNS, and large and small intestine. CNS-resident macrophages are referred to as microgial cells [32,33]. CD11b was used as the primary marker to identify myeloid cell subsets including monocytes, macrophages, granulocytes, and dendritic cells [34], although CD11b is also expressed by natural killer (NK) cells [35]. CD11b is the β subunit of the Mac-1 (CD11b/CD18) integrin that functions in myeloid cell adhesion, migration, and chemotaxis [34,35,36]. The markers that were utilized in this study to differentiate between myeloid subsets include: F4/80, Ly6C, Ly6G, and SiglecF. F4/80 is a glycoprotein with an epithelial growth factor (EGF)-like extracellular domain and a seven-transmembrane motif (TM7) that was once thought to be exclusively expressed on macrophages, but has since been shown to be expressed on eosinophils, monocytes, and Langerhans cells [24,25]. F4/80 has been theorized to play roles in regulatory T-cell differentiation and immune tolerance [37]. Ly6C is a protein that is found on 50% of bone marrow (BM) cells, but is also expressed on T-cells, NK-cells, monocytes, neutrophils, dendritic cells, and macrophages [26,38,39,40,41,42,43]. The function of Ly6C is unclear, although it has been implicated in lymphocyte differentiation, cell adhesion, cell migration, and cytokine production [38]. When paired with CD11b, Ly6C is a useful marker for distinguishing monocytes, which are split into Ly6C^−^, Ly6C^Lo^, and Ly6C^Hi^ populations for their differences in migratory and inflammatory capacity [44,45]. Ly6G is expressed almost exclusively on neutrophils [26,38,46]. The function of Ly6G is unknown, but it is suggested to be involved in myeloid expansion, cell signaling, and migration [38].

Due to high MPO expression, neutrophils served as the positive control for MPO staining in all the tissues, except the CNS, which does not have a neutrophil component in healthy animals [47]. This is shown in the spleen, whereby CD11b^+^Ly6G^+^ neutrophils (Figure 1A) expressed high levels of MPO (Figure 1B). When assessing additional CD11b^+^ subsets, we observed a population of F4/80^+^Ly6C^+/^^−^ cells in all tissues with very high expression of MPO. While teasing this population apart, we discovered that eosinophils can be identified by the low expression of Ly6C and high side scatter (SSC) [26]. Furthermore, esosinophils are identifed by SiglecF, which is a sialic acid binding immunoglobulin-like lectin [48,49]. First, due to the reported CD11b^Hi^ expression by eosinophils [50], the CD11b^Hi^ subset (Figure 1A) was analyzed for F4/80 and Ly6C expression (Figure 1C). The Ly6C^Lo^ subset was F4/80^+^ (Figure 1C) and contained a population of MPO^Hi^ cells (Figure 1D). To determine that these cells were eosinophils they were confirmed to have high SSC and express SiglecF (Figure 1E).

While eosinophil expression of MPO would be a novel finding, it is more likely that the MPO antibody utilized cross-reacted with eosinophil peroxidase (EPO), which is a heme peroxidase expressed by eosinophils that shares a large sequence homology with MPO [51,52]. As eosinophils have large granules that could exhibit autofluorescence in the FITC channel, we utilized MPO fluorescence minus one (FMO) controls to determine whether the MPO signal was an artifact. We had previously confirmed that neutrophils did not exhibit autofluorescence in the FITC channel using FMO controls (Appendix A). However, eosinophils exhibited extensive autofluorescence (Appendix A). In the histogram overlay it is clear that autofluorescence is not the sole source of the MPO^+^ signal from eosinophils (Appendix A).

The MPO antibody that was utilized here is clone 2D4, and while this is not the only commercially available antibody that is specific to MPO, when we consider the degree of homology between MPO and EPO, it is possible that other such antibodies also cross-react in a similar manner unless they were specifically tested for eosinophil reactivity. Conversely, CD11b^+^NK1.1^+^ NK-cells also appear to have an autofluorescent signal in the FITC channel (Appendix A). However, when overlayed with the FMO control, the signals overalapped indicating that the NK-cells do not express MPO (Appendix A). Neutrophils (CD11b^Hi^Ly6G^+^) and eosinophils (CD11b^+^Ly6G^−^Ly6C^Lo^SiglecF^+^SSC^Hi^) were gated out early in subsequent analyses to eliminate contamination of macrophage subsets with MPO^+^ cells.

### 2.2. MPO^+^ Macrophage Subsets in Peripheral Blood

While the peripheral blood is known to contain BM-derived monocytes, other lesser characterized macrophage populations are also present. To determine whether any peripheral blood monocyte/macrophages express MPO, CD11b^+^ cells were gated and eosinophils and neutrophils were identified using Ly6G and SiglecF (Figure 2A). Neutrophils expressing high levels of MPO are shown as the positive control (Figure 2B). While Ly6C is frequently discussed in terms of high or low expression on monocytes, there is not a cohesive agreement on how to gate these cells. Some groups distinguish between Ly6C^Med^ and Ly6C^−^ [26,45,53,54], calling only the latter Ly6C^Lo^, while other groups gate everything excluding the Ly6C^Hi^ cells as Ly6C^Lo^ [44,55,56]. Here, we have elected to keep each population separate, especially when in doing so resulted in singularly modal MPO peaks, thus presenting a more clear representation of which populations express MPO.

To specifically examine monocyte/macrophage subsets, Ly6G^−^SiglecF^−^ cells were subsequently analyzed for the expression of F4/80 and LyC, which revealed four distinct subpopulations (Subpop) (Figure 1C). Subpop I expressed the highest levels of Ly6C along with F4/80 expression (Figure 2C) and split into CD11b^Hi^ and CD11b^Lo^ subsets (Figure 2D). Interestingly, the CD11b^HI^, but not CD11b^Lo^ subset, expressed MPO (Figure 2E,F, respectively). Subpop II, which was Ly6C^Lo^F4/80^Lo/−^ (Figure 2C) also split into CD11b high and low subsets (Figure 2G), with the CD11b^Hi^, but not CD11b^Lo^ subset, expressing MPO (Figure 2H,I, respectively). Subpop III (Ly6C^−^F4/80^−^) and IV (Ly6C^−^F4/80^+^) (Figure 2C) could not be clearly separated into CD11b^Hi^ and CD11b^Lo^ subsets (data not shown). Subset III did not express MPO (Figure 2J), while Subset IV expressed low/negative levels of MPO (Figure 2K).

### 2.3. MPO^+^ Macrophage Subsets in the Spleen

The spleen is known to harbor several different macrophage populations. Among these are the red pulp macrophages, white pulp or tingible body macrophages, marginal zone macrophages (MZM), and metallophilic macrophages (MMM) [57]. White pulp macrophages are typically identified by localization within the germinal center via microscopy and described as F4/80^−^CD68^+^, thus they lack exclusive markers to identify them by flow cytometry [57,58].

To identify monocyte/macrophage subsets in the spleen, B-cells were first excluded by gating on CD19^−^ cells and then neutrophils and eosinophils were excluded before CD11b^Hi^ and CD11b^Lo^ gating on the remaining cells (Figure 3A). CD11b^Hi^ cells split into four Subpop (I-IV) when analyzed for F4/80 an Ly6C expression (Figure 3B). CD11b^Hi^ Subpop I (Ly6C^Hi^F4/80^+^) were positive for MPO (Figure 3C). The CD11b^Hi^ Subpop II (Ly6C^Lo^F4/80^+^) (Figure 3B) contained both a MPO^+^ and MPO^−^ subset (Figure 3D). In contrast, neither CD11b^HI^ Subpop III (Ly6C^−^F4/80^−^) nor IV (Ly6C^−^F4/80^+^) (Figure 3B) stained positive for MPO (Figure 3E,F, respectively). CD11b^Lo^ cells (Figure 3A) that expressed variable levels of Ly6C^Hi/−^ and were F4/80^Med/Lo^ (Figure 3G, Subpop I–IV) did not express MPO (Figure 3H–K, respectively). In contrast, CD11b^Lo^F4/80^Hi^ red pulp macrophages (Figure 3G, Subpop V), expressed low levels of MPO (Figure 3L) [59]. MPO expression in splenic monocyte/macrophages subsets is summarized in Table 1.

### 2.4. MPO^+^ Macrophage Subsets in the Central Nervous System

The primary macrophage populations in the CNS are microglial cells and perivascular macrophages. Microglial cells are considered the resident immune cell of the CNS and play important roles in CNS homeostasis and defense against pathogens [60]. Microglial cells are located throughout the entire CNS and send out processes to survey their microenvironment with the capacity to rapidly respond to stimuli leading activation that includes changes in shape, cell surface expression, and function [61]. Microglial cells are identified as CD45^Lo^CD11b^+^TMEM119^+^ [62,63]. Perivascular macrophages are found in close association with the vasculature and have been associated with a variety of diseases and are distinguished from microglial cells by a CD45^Hi^TMEM119^−^ phenotype [64,65]. Other macrophage subsets in the CNS, such as meningeal and choroid plexus macrophages, are not normally resolvable by flow cytometry and were not included in our CNS mononuclear cell preparations [66].

CNS parenchymal mononuclear cells were gated on live cells and then subsequently on CD45^+^CD11b^+^ cells (Figure 4A). The CD45^+^CD11b^+^ cells were then subgated on CD45^Lo^, CD45^Hi^, and TMEM119, identifying three separate Subpop (I–III) (Figure 4B). Gating on TMEM119 was determined through the use of an FMO control (Appendix A). When CD45^Lo^TEM119^+^ microglial cells (Figure 4B, Subpop I) were subsequently gated on F4/80 and Ly6C, three subsets were resolved (Figure 4C). Subset I.1 contained a small population of eosinophils and a second major population with low SSC (Figure 4D), which did not express MPO (Figure 4E). The latter population are likely a unique subpopulation of Ly6C^+^ microgial cells. Subset I.2 with a F4/80^Lo^Ly6C^Lo/−^ phenotype (Figure 4C) are the majority microglial cell subset, which also did not express MPO (Figure 4F). Similarly, microglial cell Subset I.3 expressing F4/80^Hi^Ly6C^−^ (Figure 4C) did not express MPO (Figure 4G). The identification of Subpop II is not clear, but is likely a TMEM119^Lo^ microglial cell subset (Figure 4B). Gating on F4/80 and Ly6C again revealed the same three subsets (Figure 4H, 1–3) as with Subpop I, all three subsets were MPO^−^ (Figure 4I–K, respectively). Subpop III are perivascular macrophages (Figure 4B) that also divided into three subsets based on F4/80 and Ly6C expression (Figure 4L, 1–3). Of the three, only Subset III.1 (F4/80^Lo^Ly6C^Hi^) contained a small number of cells that may express a small level of MPO (Figure 4M–O, respectively).

### 2.5. MPO^+^ Macrophage Subsets in the Gut

The gastrointestinal tract contains a number of macrophage subsets that are important in gut homeostasis and recognition of pathogens, some of which are regionally localized. We separated the gut into the small and large intestine. We also examined the MLN. Although there are a multitude of markers that have been used to distinguish macrophages in the gut, we opted to utilize a relatively simple flow cytometry panel that would still allow the identification of specific subsets that express MPO.

Large intestinal cells were gated on CD45^+^CD11b^+^ and neutrophils and eosinophils were excluded by Ly6G and SiglecF expression (Figure 5A). Neutrophils served as the positive control for MPO expression (Figure 5B). Subsequent gating on F4/80 and Ly6C revealed four distinguishable Subpop (Figure 5C, I–IV). Subpop I (F4/80^Hi^Ly6C^HI^) split into CD11b^Hi^ (Figure 5D, Subset I.1) and CD11b^Lo^ (Figure 5D, Subset I.2) subsets, of which only I.1 expressed MPO (Figure 5E,F, respectrively). Subpop II–IV were not further differentiated by CD11b expression. Subpop II (F4/80^Lo/−^Ly6C^Lo^) (Figure 5C) split into MPO-negative and -positive subsets (Figure 5G). Subpop III (F4/80^−^Ly6C^−^) (Figure 5C) did not express MPO (Figure 5H). Interestingly, Subpop IV (F4/80^+^Ly6C^−^) (Figure 5C) expressed MPO at a low level (Figure 5I).

In isolating the small intestine Peyer’s patches were not examined separately, thus our analysis is a collection of total macrophage populations. The small intestine was also gated on CD45^+^CD11b^+^ cells with subsequent analysis revealing eosinophil and neutrophil subsets (Figure 6A), the latter of which served as the positive control for MPO (Figure 6B). When SiglecF^−^Ly6G^−^ cells were analyzed for F4/80 and Ly6C expression, five separate Subpop (I–V) were observed (Figure 6C). As in the large intestine, Subpop I but not II–V, was further subdivided by CD11b expression (Figure 6D) and Subset I.1, but not I.2, was MPO-positive (Figure 6E,F, respectively). Subpop II–IV in terms of F4/80, Ly6C, and MPO expression were identical to the large intestine (Figure 6C,G–I, respectively). Subpop V (F4/80^+^Ly6C^Lo^) (Figure 6C) is unique to the small intestine and expressed a high level of MPO (Figure 6J) that was similar to neutrophils in the blood and spleen (Figure 1B and Figure 2B).

CD64 is the high affinity IgG receptor FcγR, a marker that **is** commonly used to distinguish gut macrophages from dendritic cells [67]. Subset IV (Figure 5C and Figure 6C) in both the large and small intestine were predominantly CD64^+^, which was not observed in any of the other subsets (data not shown).

MLN were analyzed similar to the intestines, gating on CD45^+^CD11b^+^ cells in which neutrophils and eosinophils were identified using Ly6G and SiglecF, respectively (Figure 7A). Interestingly, most of the Ly6G^+^ cells (Figure 7A) did not express MPO (Figure 7B). The separation of macrophages by F4/80 and Ly6C resulted in three Subpop (I-II) largely based on the differential expression of Ly6C (Figure 7C). As with the small and large intestine, Subpop I was further divided by CD11b (Figure 7D), with the CD11b^+^ (Figure 7E), but not CD11b^−^ (Figure 7F), expressing MPO. Again, Subpop II (Figure 7C) contained an MPO^+^ subset (Figure 7G). Subpop III (Figure 7C) that contained F4/80^Lo/−^ cells did not express MPO (Figure 7H). MPO expression in gut monocyte/macrophage subsets is summarized in Table 1.

## 3. Discussion

By performing a comprehensive flow cytometric analysis of monocyte/macrophage populations in multiple tissues, the data show that only a subset of them express MPO. Any given tissue can be populated by diverse subsets of monocytes/macrophages thus, we chose a phenotyping strategy that would be applicable to all tissues that were examined. As tissue digestion leads to the release of many cell types, CD45 was used for the identification of hematopoietic cells. We chose to gate on CD11b because it is considered to be a pan marker of the myeloid lineage. Neutrophils were eliminated by Ly6G expression and eosinophils by SiglecF [26,68]. F4/80 was chosen because it has been used as a marker of monocyte/macrophage subsets in all tissues examined, i.e., expressed by both circulating monocytes and tissue-resident macrophages [24,25,26,69,70,71]. Ly6C was used because it is known to have differential expression on monocyte/macrophage subsets and can be used to differentiate between monocyte and macrophage subsets [26,72,73,74]. All MPO^+^ monocyte/macrophage populations expressed either F4/80 and/or Ly6C, regardless of the tissue that was examined. In addition, every tissue except the CNS, contained at least one monocyte/macrophage subset with medium-high MPO expression. These data demonstrate that MPO can be used as an additional marker to differentiate between the plethora of monocyte/macrophage subsets.

Monocytes/macrophages are highly dynamic cells that are able to quickly adapt to their environment thereby performing specific functions in a tissue-specific manner. All tissues have macrophage populations that are essential for tissue homeostasis, salvaging of dead and dying cells, and the detection of micro-organisms as part of the innate immune system. In the tissues that we examined, monocytes/macrophages that maintain the endothelium in the blood play a role in immune cell turnover and contribute to adaptive immunity in the spleen, regulate neuronal activity at the synapse in the CNS, and interact with and regulate the gut microbiome and maintain intestinal homeostasis in the gut [33,71,75,76]. There are two sources of monocytes that either circulate in the peripheral blood and lymphatics or enter tissues. The first is the yolk sac/fetal liver that fate mapping studies have shown generate a multitude of tissue-specific macrophages that remain for life including CNS microglial cells, liver Kupffer cells, and lung alveolar macrophages [77]. In other tissues such as the spleen and gut, at least some embryonic-derived macrophage subsets are replaced over time by BM-derived adult macrophages [77,78]. Ly6C^+^ tissue macrophages are thought to be derived from the circulation and have pro-inflammatory function [41]. Ly6C^−^ macrophages are regarded as tissue-specific macrophages of embryonic origin [55].

The peripheral blood is the conduit for immune cells that traffic around the body for immune surveillance and to perform homeostatic functions. Based on the literature, CD11b^+^Ly6C^+^ cells in the blood are BM-derived monocytes, which is Subpop I in Figure 2C [44,45,79]. Subpop I split into CD11b^Hi^ and CD11b^Lo^ subsets (Figure 2D). The CD11b^Hi^ subset expressed MPO (Figure 2E), which is consistent with their pro-inflammatory functions. Similarly, in Ly6C^Lo^ cells (Figure 2C, Subpop II), the CD11b^Hi^ cells expressed MPO (Figure 2H). In contrast, in CD11b^Lo^ cells neither the Ly6C^Hi^ nor Ly6C^Lo^ subsets expressed MPO (Figure 2F,I). The origin of these cells is unclear, but their presence in the gut (Figure 5, Figure 6 and Figure 7F, but not spleen, suggest they are migratory tissue macrophages. While both Ly6C^Hi^ and Ly6C^Lo^ blood monocytes have demonstrated similar phagocytic capacity, only the Ly6C^Hi^ subset expressed high levels of CCR2 and CD62L and are thus theorized to play an inflammatory role when they are recruited to sites of infection [44,45], which is consistent with MPO expression. Ly6C^Lo^ monocytes are called patrolling monocytes, which crawl along blood vessels where they conduct immune surveillance for the surrounding tissues and are important for tissue repair [80]. In addition, Ly6C^Lo^ monocytes were shown to be anti-inflammatory and enter the tissues under homeostatic conditions [45]. These latter two findings are consistent with the lack of MPO expression.

Ly6C^Hi^ blood monocytes give rise to Ly6C^Lo^ monocytes [15,44,45,79]. In the blood Ly6C^Hi^ monocytes express more MPO than Ly6C^Lo/−^ cells (Figure 2E versus Figure 2H), implying greater inflammatory capacity for these less mature cells. This observation is further supported by the literature which suggests that “classical” Ly6C^Med/Hi^ monocytes/macrophages are inflammatory in nature, while “non-classical” Ly6C^Lo/−^ monocytes/macrophages tend to be anti-inflammatory and, in some cases, regenerative [15,45]. The Ly6C^−^ subset split into F4/80^−^MPO^−^ and F4/80^Lo^MPO^+/−^ subsets (Figure 2C, Subsets III and IV, respectively). The origin of the Ly6C^−^F4/80^−^ cells is not clear, but they are likely patrolling monocytes that are MPO^−^ (Figure 2J). This subset could also contain a small subset of NK-cells that express CD11b due to the light scatter gating including some large granular lymphocytes [81]. Interestingly, the Ly6C^−^F4/80^Lo^ subset (Figure 2C, Subset IV) split into MPO^+^ and MPO^−^ populations (Figure 2K). The lack of Ly6C expression suggests that they are circulating/migrating macrophage populations, but their tissues of origin are unknown.

Other markers to differentiate blood monocytes are the chemokine receptors CX3CR1 and CCR2. CX3CR1^Hi^CCR2^Lo^ monocytes (Ly6C^Lo/−^) remain in the blood [80]. CX3CR1^Lo^CCR2^Hi^ are inflammatory monocytes (Ly6C^Hi^) that enter tissues, promote inflammation, and exhibit anti-microbial functions, which is consistent with their MPO expression [44,45,80].

The spleen is known to harbor a number of macrophages with specific functions and localization and also contains circulating subsets. These include circulating monocytes which comprise ~50% of the CD11b^Hi^ subset (Figure 3A,B (Subpop I)). Subpop III (Figure 3B) could also be from the circulation, but also matches the expression pattern of a novel dendritic cell-like called L-DC that were characterized as CD11b^Hi^CD11c^Lo^MHCII^−^/CD43^+^Ly6C^−^Ly6G^−^SiglecF^−^ [82,83]. Further characterization using CD11c, MHC Class II, and CD43 is required to confirm, but a F4/80^−^ expression pattern is consistent with population III not being a monocyte/macrophage.

The splenic marginal zone (MZ) is at the interface of the non-lymphoid red pulp that is rich in erythrocytes and the lymphoid white pulp. Blood from the circulation flows through the MZ making it an important first-line of defense against blood-borne pathogens [84]. MMM reside on the white pulp side of the sinus and MZM on the red pulp side allowing their identification by histology [84]. To resolve them by flow cytometry required digestion in a three enzyme cocktail, which we also utilized [85]. Both populations are highly phagocytic with a dependence upon the nuclear receptor LXR1α for promotion of the phagocytic cascade while maintaining an anti-inflammatory phenotype allowing the removal of cells without inducing an innate immune response [86,87,88]. Both MZ macrophage subsets also fall within the CD11b^Hi^ gate (Figure 3A) and are F4/80^Med^Ly6C^Lo/−,^ indicating they are likely Subpop II and/or IV in Figure 3B [85]. Subpop IV is MPO^−^ (Figure 3F), while Subpop II contains an MPO^−^ and MPO^+^ subset (Figure 3D). Despite the use of enzymatic digestion of the spleen [26,85] and the utilization of reported markers such as MARCO, CD169, and Tim4 [57,58,85,89], we were unable to differentiate MZ and MMM by flow cytometry. We attribute this to the proteolytic removal of some of the identifier proteins from the cell surface, but this was not confirmed.

The splenic CD11b^Lo^ macrophage subsets (Figure 3B) were similar to CD11b^Hi^ in terms of F4/80 and Ly6C expression (Figure 3B versus Figure 3G). Based on MPO expression, CD11b^Lo^ Subpop I is not composed of circulating monocytes (Figure 3C versus Figure 3H) and Subpop II lacks the MPO^+^ subset that was seen in the CD11b^Hi^ fraction (Figure 3D versus Figure 3I). Subpop III is MPO^−^ in both CD11b subsets (Figure 3E versus Figure 3J). Although F4/80 is considered a monocyte/macrophage marker, its expression has been shown on splenic dendritic cells with CD8^+^F4/80^+^ and CD8^−^F4/80^+/−^ phenotypes [90]. In addition, both Ly6C^Hi^ and Ly6C^Lo^ monocytes can differentiate into dendritic cells [42]. Thus, it is possible that all of the MPO^−^ subsets represent various classical splenic dendritic cell subsets, some of which express CD11b [91]. The inclusion of CD11c staining can be used to confirm a dendritic cell identification [91]. Similar MPO^−^ subsets, likely of dendritic cell origin, were found in the spleen and gut tissues.

One unique CD11b^Lo^ splenic Subpop was identified (Figure 3G, Subpop V), that are likely red pulp macrophages based on the F4/80^Hi^CD11b^Lo^ expression pattern [92]. Red pulp macrophages are derived from the fetal liver [92]. Their primary role is to recycle injured and senescent erythrocytes thereby playing a role in iron metabolism [93]. They also remove blood-borne particulates [93]. Red pulp macrophages can induce regulatory T-cells via the production of TGF-β and participate in parasitic infections via the production of Type I interferons [93]. Interestingly, red pulp macrophages split into an MPO^−^ and MPO^Lo^ subset (Figure 3L). VCAM-1 and CD68 expression can be used to more conclusively identify if red pulp macrophages express MPO [85,94]. Consistent with their role of homeostasis, the low/negative expression of MPO indicates they are not innately pro-inflammatory.

Although MPO expression by microglial cells in the CNS has been reported histologically, the studies were not performed in a manner that allowed the specific identification of microglial cells. In addition, these studies did not differentiate between the CNS-resident and infiltrating macrophages. Most notably, these studies were performed in the autoimmune disease multiple sclerosis and its animal model experimental autoimmune encephalomyelitis [17,95,96]. Our studies clearly show that microglial cells do not express MPO in the steady state (Figure 4B–K). This is consistent with a study that also utilized flow cytometry to show that MPO^+^ cells in EAE were neutrophils and Ly6C^hi^ macrophages, but not microglial cells [97]. In general, perivascular macrophages also did not express MPO (Figure 4B,L,N,O), with the exception of a small subset of Ly6C^+^ cells (Figure 4M). It is not clear whether these cells are resident to the CNS or blood-derived cells that were not removed by perfusion. These results indicate that while microglial cells and perivascular macrophages are the innate immune sentinels of the CNS, they likely do not directly participate in microbial killing, at least via MPO. Pro-inflammatory functions are known to be mediated by infiltrating macrophages in a variety of CNS disorders and infections [98,99].

Although the gastrointestinal tract spans from the mouth to the anus, here we concentrated on the macrophage populations in the large and small intestine, because that is the primary location of gut-associated lymphoid tissues (GALT). GALT structures in the small intestine include Peyer’s patches and isolated lymphoid follicles that are found in the large intestine [100,101,102]. Immune cells are also located in the lamina propria, a thin layer of connective tissue just under the epithelium that is present in both the small and large intestines [103,104]. In addition, both the large and small intestines drain to the MLN, but not to the same nodes [105]. Here, we opted to analyze the large and small intestines separately without separating the individual GALT structures. Similarly, all nodes of the MLN were combined. This provided a snapshot of whether MPO could be used in downstream studies to identify specific macrophage subsets, which have been difficult to differentiate [71,104].

Interestingly, the MPO expression patterns in the large and small intestines, as well as the MLN, were relatively similar, with similarities to the peripheral blood and spleen. CD11b^Hi^ cells from Subpop I, representing circulating inflammatory monocytes, expressed MPO, while the CD11b^Lo^ subset did not (Figure 5C–F, Figure 6C–F and Figure 7C–F). Subpop II contained both MPO^+^ and MPO^−^ cells in the intestines, but not MLN (Figure 5C,G, Figure 6C,G and Figure 7C,G). Subpop III was MPO^−^ (Figure 5C,H, Figure 6C,H and Figure 7C,H). Subpop IV expressed MPO in the large and small intestine, but was not present in the MLN (Figure 5C,I, Figure 6C,I and Figure 7C,I). Collectively, these data indicate that blood monocytes circulate freely through GALT tissues and are the only MPO^+^ monocyte/macrophage in the MLN.

Subpop IV in the large intestine had a bimodal MPO peak with a low and high MPO subset (Figure 5C,I). The MPO^Hi^ subset was clearly distinguishable in the small intestine, which was designated Subset V (Figure 6C,J). This subset had similar levels of F4/80 as the circulating blood monocytes, but in contrast were Ly6C^Lo^, similar to patrolling monocytes (Figure 2C versus Figure 6C). Thus, their identification is not clear, but due to the high MPO levels, likely play a role in clearing bacteria that disseminate across the gut epithelial barrier. This function can be performed by lamina propria macrophages which are CX3CR1^Hi^ [71]. Fate mapping studies have shown that gut macrophages are derived from hematopoietic BM stem cells and originate from circulating Ly6C^Hi^, but not Ly6C^Lo^ blood monocytes [71,78]. Monocyte to macrophage differentiation in the lamina propria has been termed the monocyte “waterfall” that progresses through various stages, culminating in cells that upregulate MHC Class II, downregulate Ly6C, and acquire CX3CR1 expression [71].

In examining MPO expression levels in neutrophils, those in the gut expressed ~3-fold lower MPO levels as the peripheral blood, comparing MFI (Figure 2B, Figure 5B and Figure 6B). Blood neutrophils will have recently been released from the BM due to their short-lived half-life of 6–8 h [106]. Neutrophils circulate through tissues in the steady state including the intestines, where they are important for homeostasis [107]. Neutrophils eliminate commensals that cross the epithelial barrier by a variety of mechanisms including MPO. Neutrophils can also cross the epithelial barrier and enter the lumen [108]. MPO is stored in azurophilic granules, which can be released though the process of degranulation [6,109,110]. MPO utilizes H_2_O_2_ to generate hyphochlorous acid, which is a potent microbicidal [111,112]. The reduced levels of MPO expression in intestinal neutrophils indicates that they actively undergo degranulation in the steady state. Of particular interest is that MLN neutrophils that were identified by Ly6G expression (Figure 7A) were largely negative for MPO (Figure 7B). This could reflect migration of degranulated neutrophils out of the gut and into the lymphatics or that an unidentified Ly6G^+^ cell is present in the MLN.

Here, we investigated whether MPO could be used as an additional marker to distinguish the multitude of monocyte/macrophage subsets in the circulation and tissues. The tissues/organs that we chose to examine were of interest to our research but the markers utilized can be applied to any tissue or organ. Our studies clearly show that both MPO^−^ and MPO^+^ monocyte/macrophage subsets exist. Most notably, circulating blood monocytes that originate from the BM express intermediate levels of MPO. Of particular interest to us [6,18] is our finding that CNS resident microglial cells and perivascular macrophages do not express MPO, contrary to other reports [17,95,96]. Overall, the trends were that all MPO^+^ monocytes/macrophages expressed F4/80 and/or Ly6C, with CD11b^Hi^F4/80^Hi^Ly6C^+^ subsets expressing the highest levels of MPO. These collective data demonstrate that MPO expression can be used as an additional marker of monocyte/macrophages and when combined with other markers can provide information on subset function and origins.

## 4. Materials and Methods

### 4.1. Mice

B10.PL mice were purchased from The Jackson Laboratories (Bar Harbor, ME, USA) and housed and bred in the Translational Biomedical Research Center of the Medical College of Wisconsin (MCW). Animal protocols using all the relevant ethical regulations were approved by the MCW Institutional Animal Use and Care Committee. The mice that were used were 6-8 weeks of age and were female.

### 4.2. Antibodies and General Methods

All antibodies utilized were anti-mouse. CD45.2-Pacific Blue, CD45.2-Alexa Fluor 700, CD11b-PE/Cy7, F4/80-PE/Dazzle594, Ly6C-PE, and Ly6G-PE/Cy7were purchased from Biolegend (San Diego, CA, USA). CD11b PerCP-Cy5.5 was purchased from BD Biosciences (San Jose, CA, USA). MPO-FITC was purchased from LSBio (Seattle, WA, USA). TMEM119-Alexa Fluor 647 was purchased from Abcam (Cambridge, England, UK). The anti-CD16/CD32 2.4G2 mAb was grown locally and used as a mouse Fc block. All immune cell isolations were performed using polypropylene plasticware and the cell suspensions were kept on ice whenever possible.

### 4.3. Isolation of Splenic Macrophages

Mouse spleens were minced into a homogenous pulp with a scalpel before suspension into DMEM (Gibco, Amarillo, TX, USA) containing 0.1 mg/mL DNase I (Roche, Indianapolis, IN, USA), 0.5 U/mL Dispase (Stemcell Technologies, Vancouver, BC, CAN), 1 mg/mL Collagenase D (Roche), and 2% FBS. Spleen homogenates were incubated at 37 °C for 30 min before passed through a 70 µM cell strainer. The cell suspension was washed with 4 volumes of DMEM and centrifuged at 500× *g* for 5 min before re-suspension into ACK lysis buffer (150 mM ammonium chloride, 10 mM potassium bicarbonate, 0.1 mM EDTA) for 10 min at room temperature, followed by one wash with cold DMEM before flow cytometry staining.

### 4.4. Isolation of CNS Mononuclear Cells

Mice were anesthetized with a cocktail containing ketamine and xylazine prior to perfusion with 20 mL of cold phosphate-buffered saline (PBS) injected into the left ventricle of the heart before dissection. Brains and spinal cords were extracted and placed in cold Hank’s Balanced Salt Solution (HBSS) without calcium or magnesium (Gibco). The CNS tissue was finely minced with a scalpel prior to digestion in HBSS containing 2 mg/mL collagenase D (Roche) and 14 µg/mL DNase I (Roche) and incubated for 15 min at 37 °C before the reaction was stopped with the addition of 4 volumes of cold HBSS. The cell suspension was passed through a 70 µM cell strainer and washed with 45 mL HBSS. The cells were centrifuged at 500× *g* for 7 min at 4 °C before re-suspension into 10 mL of 37% Percoll (Sigma, St Louis, MO, USA) and centrifugation at 500× *g* for 10 min without braking. The lipid layer was aspirated off and the cell pellet was collected and re-suspended in DMEM before flow cytometry staining.

### 4.5. Isolation of Gut Cells

The large and small intestines were collected in PBS with penicillin/streptomycin (Gibco). Excess fatty tissue was removed from the intestines, which were then cut longitudinally and cut again into 2 cm strips. The intestines were washed by vigorous shaking, and replaced with fresh PBS three times. The intestine pieces were then incubated in RPMI 1640 containing 10 mM HEPES (Gibco), 25 mM sodium bicarbonate (Corning, NY, USA), 5 mM EDTA (Fisher, Waltham, MA, USA), 5 mM 1,4-Dithiothreitol (DTT) (Sigma), and 2% FBS at 37 °C for 20 min. This was repeated for another 20 min before rinsing the tissue pieces with fresh RPMI and cutting them into smaller 0.5 cm pieces, which were incubated in 10 mL of RPMI containing 10 mM HEPES, 25 mM sodium bicarbonate, 0.1 mg/mL Liberase (Roche), 0.5 U/mL Dispase (Roche), 0.1 µg/mL DNase I (Sigma), and 10% FBS at 37 °C for 45 min. The remaining tissue pieces were briefly triturated before the cell suspension was filtered through a 70 µM cell strainer and washed with 20 mL fresh RPMI. The cells were centrifuged at 400× *g* 4 °C for 10 min before re-suspension into 10 mL RPMI for counting and flow cytometry staining.

### 4.6. Isolation of Peripheral Blood Lymphocytes

Mice were bled from the submental space into 3.8% sodium citrate. ACK lysis buffer was added to the blood which was then incubated for 10 min at room temperature. The cells were washed twice with PBS before proceeding with staining for flow cytometry.

### 4.7. Flow Cytometry

A total of 1–2 × 10^6^ cells were stained with Zombie Violet viability dye (Biolegend, San Diego, CA, USA) according to the manufacturer specifications and then washed with 500 μL FACS buffer (PBS, 2% FBS, 0.1% sodium azide) before a 10 min incubation with 0.5 µg FcR blocking antibody (2.4G2). The cells were washed with FACS buffer and incubated with a surface antibody cocktail for 15 min on ice. The cells were washed twice before staining intracellularly for MPO using the IC fixation buffer kit from eBioscience (San Diego, CA, USA) in accordance with manufacturer instructions. The cells were re-suspended into FACS buffer and data were acquired from the live cells on a BD LSRII flow cytometer. The data were analyzed with FlowJo software (FlowJo, Ashton, OR, USA).

## 5. Conclusions

Here, we utilized monocyte/macrophage cell surface markers standard to the field in combination with the intracellular protein MPO to identify subsets with proinflammatory potential. All tissues examined, except the CNS, contained at least one subset of MPO^+^ monocyte/macrophages. The data presented provides evidence that many of these subsets originate from blood BM-derived monocytes with high MPO expression. This supports the role of blood monocytes as an early defense mechanism against invading microorganisms.

## Figures and Tables

**Figure 1 ijms-23-08246-f001:**
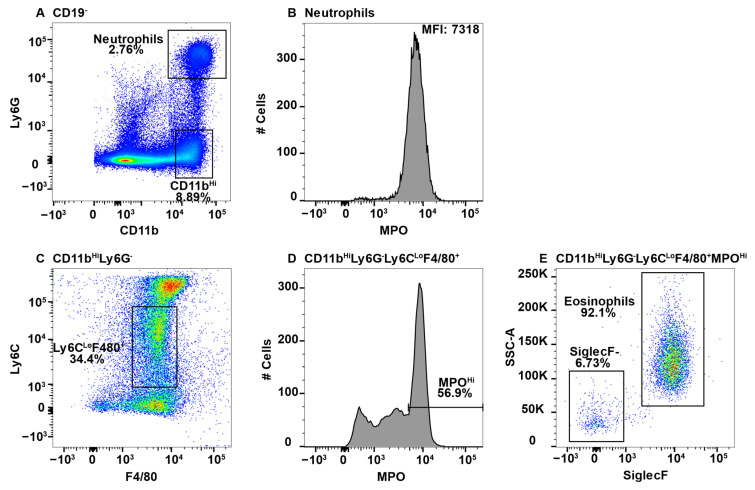
Using neutrophils as a positive control for MPO expression by flow cytometry, eosinophils were found to give a false positive signal. Splenocytes were stained with antibodies specific for CD19, CD11b, Ly6G, F4/80, Ly6C, MPO, and SiglecF and using flow cytometry, the CD19^−^ cells were gated out. CD11b^+^Ly6G^+^ neutrophils (**A**) were analyzed for MPO expression (**B**). CD11b^Hi^Ly6G^−^ cells were analyzed for F4/80 and Ly6C expression (**C**) and F4/80^+^Ly6C^Lo^ MPO expression is shown (**D**). MPO^Hi^ cells were further analyzed for SSC^Hi^ and SiglecF expression to identify eosinophils (**E**). The percent positive cells in dot plot gates and the MFI of histograms is provided. Data shown are representative of three mice.

**Figure 2 ijms-23-08246-f002:**
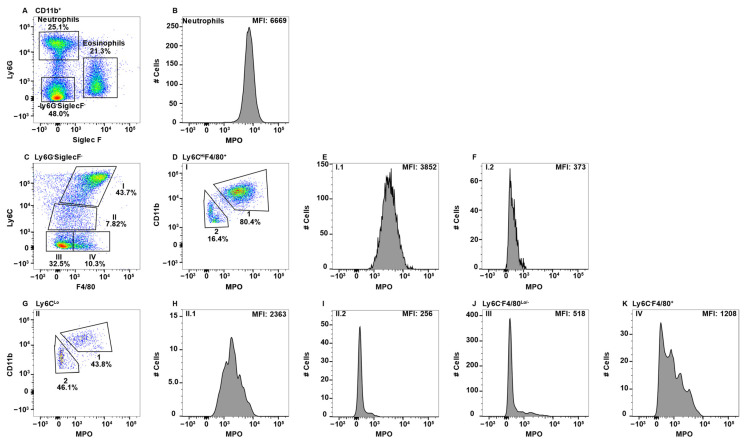
Blood monocyte/macrophage subsets differentially express MPO. PBMC were stained with antibodies specific for CD45, CD11b, Ly6C, Ly6G, F4/80, MPO, and SiglecF and using flow cytometry, CD45^+^ cells were gated on CD11b. Neutrophils and eosinophils were identified by expression of Ly6G and SiglecF, respectively (**A**). Neutrophil MPO positive staining is shown (**B**). CD11b^+^Ly6G^−^SiglecF^−^ cells were analyzed for F4/80 and Ly6C expression and four Subpop (I–IV) were identified (**C**). Subpop I F4/80^+^Ly6C^Hi^ cells were separated into CD11b^Hi^ (1) and CD11b^Lo^ (2) populations (**D**), and MPO expression is shown (**E** and **F**, respectively). Subpop II Ly6C^Lo^ cells were separated into CD11b^Hi^ (1) and CD11b^Lo^ (2) populations (**G**), and MPO expression is shown (**H** and **I**, respectively). MPO expression in Subpop III F4/80^−^Ly6C^−^ cells (**J**) and Subpop IV F4/80^+^Ly6C^−^ cells (**K**) is shown. The percent positive cells in dot plot gates and the MFI of histograms is provided. Data shown are representative of three mice.

**Figure 3 ijms-23-08246-f003:**
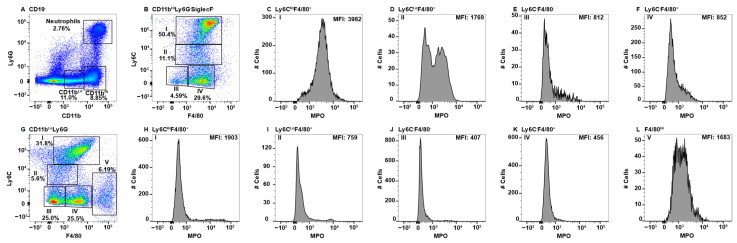
Splenic monocyte/macrophage subsets differentially express MPO. Splenocytes were stained with antibodies specific for CD19, CD11b, Ly6C, Ly6G, F4/80, MPO, and SiglecF and using flow cytometry, CD19^−^ cells were analyzed for Ly6G and CD11b to identify double positive neutrophils and Ly6G^−^CD11b^Hi^ and Ly6G^−^CD11b^Lo^ cells (**A**). CD11b^Hi^Ly6G^−^SiglecF^−^ cells were analyzed for F4/80 and Ly6C expression and four Subpop (I–IV) were identified (**B**). MPO expression in Subpop I F4/80^+^Ly6C^Hi^ (**C**), II Ly6C^Lo^F4/80^+^ (**D**), III Ly6C^−^F4/80^−^ (**E**), I and IV Ly6C^−^F4/80^+^ (**F**) is shown. CD11b^Lo^Ly6G^−^SiglecF^−^ cells were analyzed for F4/80 and Ly6C expression and five Subpop (I–V) were identified (**G**). MPO expression in Subpop I F4/80^+^Ly6C^Hi^ (**H**), II Ly6C^Lo^F4/80^+^ (**I**), III Ly6C^−^F4/80^−^ (**J**), IV Ly6C^−^F4/80^+^ (**K**), and V Ly6C^−^F4/80^Hi^ (**L**) is shown. The percent positive cells in dot plot gates and the MFI of histograms is provided. Data shown are representative of three mice.

**Figure 4 ijms-23-08246-f004:**
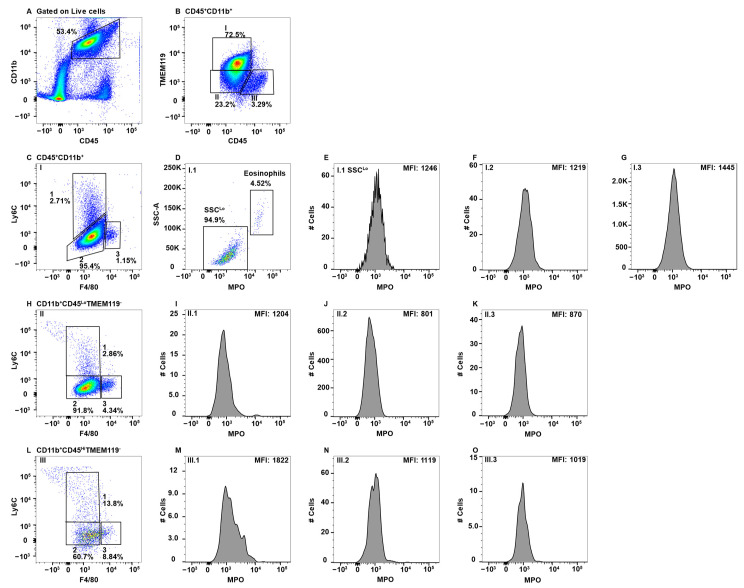
CNS microglial cells and perivascular macrophages do not express MPO. Mononuclear cells from brains and spinal cords were stained with antibodies specific for CD45, CD11b, Ly6C, F4/80, MPO, and TMEM119 and using flow cytometry cells were gated on CD45^+^CD11b^+^ (**A**), which were analyzed for TMEM119 expression identifying three Subpop (I-III) (**B**). Subpop I CD45^Lo^CD11b^+^TMEM119^+^ cells were analyzed for Ly6C and F4/80 expression, which identified three subsets (I.1, I.2, and I.3) (**C**). SSC^HI^ eosinophils were eliminated in Subset I.1 (**D**) prior to analysis of MPO expression (**E**). Subsets I.2 and I.3 did not require eosinophil exclusion before MPO analysis (**F** and **G**, respectively). Subpop II CD45^Lo^CD11b^+^TMEM119^−^ cells were analyzed for Ly6C and F4/80 expression (**H**), which identified three subsets (II.1, II.2, and II.3), which were further analyzed for MPO expression (**I**,**J**,**K**, respectively). Subpop III CD45^Hi^CD11b^+^TMEM119^−^ cells analyzed for Ly6C and F4/80 expression identified three subsets (III.1, III.2, and III.3) (**L**), which were further analyzed for MPO expression (**M**,**N**,**O**, respectively). The percent positive cells in dot plot gates and the MFI of histograms is provided. Data shown are representative of three mice.

**Figure 5 ijms-23-08246-f005:**
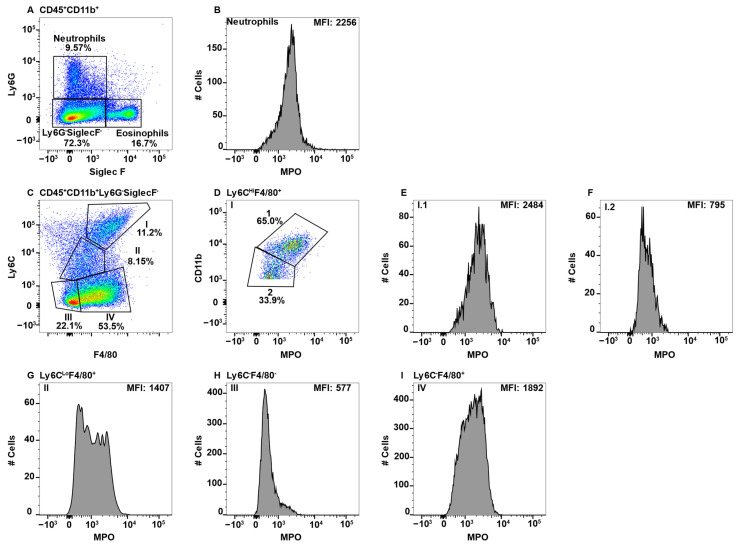
Large intestinal macrophage subsets differentially express MPO. Large intestine mononuclear cells were stained with antibodies specific for CD45, CD11b, Ly6C, Ly6G, F4/80, MPO, and SiglecF and using flow cytometry, CD45^+^CD11b^+^ cells were analyzed for Ly6G and SiglecF expression to identify Ly6G^+^SiglecF^−^ neutrophils and Ly6G^−^SiglecF^+^ eosinophils (**A**). Neutrophil MPO-positive staining is shown (**B**). CD45^+^CD11b^+^Ly6G^−^SiglecF^−^ cells analyzed for F4/80 and Ly6C identified four Subpop (I–IV) (**C**). Subpop I F4/80^+^Ly6C^Hi^ cells were separated into CD11b^Hi^ (1) and CD11b^Lo^ (2) populations (**D**), and MPO expression is shown (**E** and **F**, respectively). MPO expression in Subpop II Ly6C^Lo^F4/80^+^ (**G**), III Ly6C^−^F4/80^−^ (**H**) and IV Ly6C^−^F4/80^+^ (**I**) is shown. The percent positive cells in dot plot gates and the MFI of histograms is provided. Data shown are representative of three mice.

**Figure 6 ijms-23-08246-f006:**
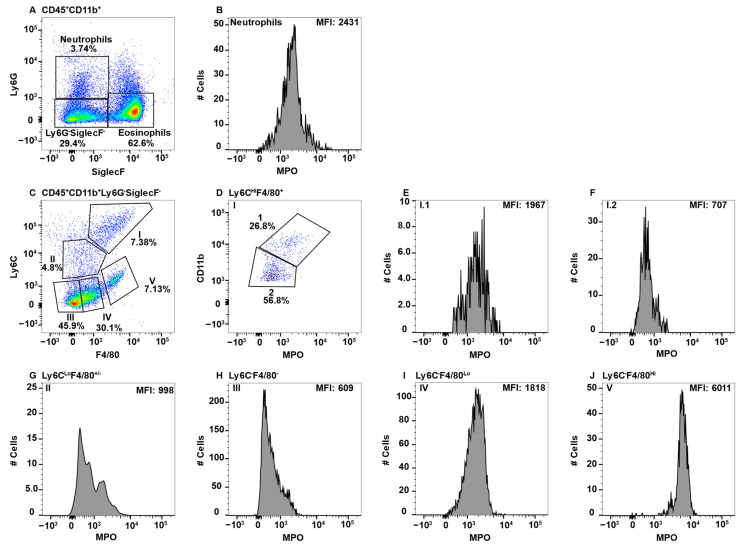
Small intestinal macrophage subsets differentially express MPO. Small intestine mononuclear cells were stained with antibodies specific for CD45, CD11b, Ly6C, Ly6G, F4/80, MPO, and SiglecF and using flow cytometry, CD45^+^CD11b^+^ cells were analyzed for Ly6G and SiglecF expression to identify Ly6G^+^SiglecF^−^ neutrophils and Ly6G^−^SiglecF^+^ eosinophils (**A**). Neutrophil MPO-positive staining is shown (**B**). CD11b^Hi^Ly6G^−^SiglecF^−^ cells were analyzed for F4/80 and Ly6C expression and five Subpop (I–V) were identified (**C**). Subpop I F4/80^+^Ly6C^Hi^ cells were separated into CD11b^Hi^ (1) and CD11b^Lo^ (2) populations (**D**), and MPO expression is shown (**E** and **F**, respectively). MPO expression in Subpop II Ly6C^Lo^F4/80^+^ (**G**), III Ly6C^−^F4/80^−^ (**H**), IV Ly6C^−^F4/80^+^ (**I**), and Ly6C^−^F4/80^Hi^ V (**J**) is shown. The percent positive cells in dot plot gates and the MFI of histograms is provided. Data shown are representative of three mice.

**Figure 7 ijms-23-08246-f007:**
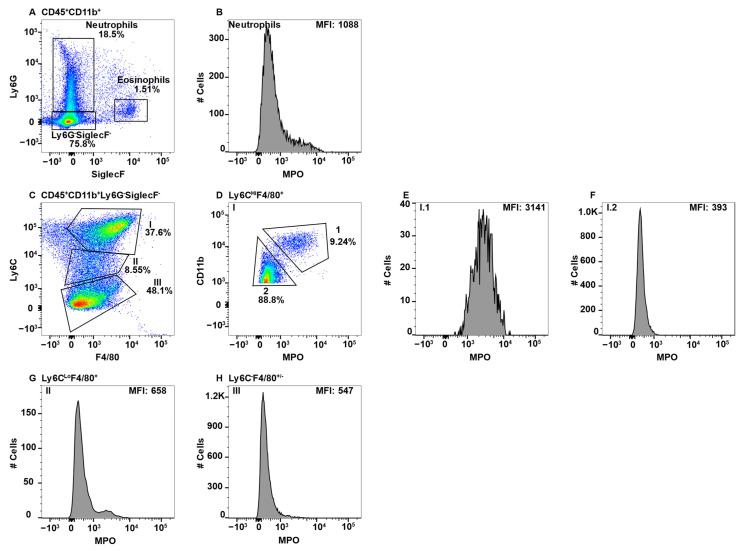
MLN macrophage subsets differentially express MPO. MLN cells were stained with antibodies that were specific for CD45, CD11b, Ly6C, Ly6G, F4/80, MPO, and SiglecF and using flow cytometry CD45^+^CD11b^+^ cells were analyzed for Ly6G and SiglecF expression to identify Ly6G^+^SiglecF^−^ neutrophils and Ly6G^−^SiglecF^+^ eosinophils (**A**). Neutrophil expression of MPO is shown (**B**). CD45^+^CD11b^−^Ly6G^−^SiglecF^−^ cells were analyzed for F4/80 and Ly6C expression and three Subpop (III) were identified (**C**). Subpop I F4/80^+^Ly6C^Hi^ cells were separated into CD11b^Hi^ (1) and CD11b^Lo^ (2) populations (**D**), and MPO expression is shown (**E** and **F**, respectively). MPO expression in Subpop II Ly6C^Lo^F4/80^+^ (**G**) and III Ly6C^−^F4/80^−^ (**H**) is shown. The percent positive cells in dot plot gates and the MFI of histograms is provided. Data shown are representative of three mice.

**Table 1 ijms-23-08246-t001:** MPO Expression Level in Monocyte/macrophage Subsets Differentiated by Ly6C, F4/80 and CD11b Cell Surface Expression.

**Blood ^1^**		**Spleen ^2^** **CD11b^Hi^**		**Spleen ^2^** **CD11b^Lo^**		**LI ^3^**		**SI ^4^**		**MLN ^5^**	
**Subset**	**MPO ^6^/Predicted Identity**	**Subset**	**MPO/Predicted Identity**	**Subset**	**MPO/Predicted Identity**	**Subset**	**MPO/Predicted Identity**	**Subset**	**MPO/Predicted Identity**	**Subset**	**MPO/Predicted Identity**
**I.1** Ly6C^Hi^ F4/80^+^ CD11b^Hi^	+++ BM-derived monocyte	**I** Ly6C^Hi^ F4/80^+^	+++ BM-derived monocyte	**I** Ly6C^Hi^ F4/80^+^	− Dendritic Cell	**I.1** Ly6C^Hi^ F4/80^+^ CD11b^Hi^	++ BM-derived monocyte	**I.1** Ly6C^Hi^ F4/80^+^ CD11b^Hi^	+ BM-derived monocyte	**I.1** Ly6C^Hi^ F4/80^+^ CD11b^+^	+++ BM-derived monocyte
**I.2** Ly6C^Hi^ F4/80^+^ CD11b^Lo^	− Migrating Tissue Macro ^7^	**II** Ly6C^Lo^ F4/80^+^	− & ++ MZ or MMM Macro	**II** Ly6C^Lo^ F4/80^+^	− Dendritic Cell	**I.2** Ly6C^Hi^ F4/80^+^ CD11b^Lo^	− Migrating Tissue Macro	**I.2** Ly6C^Hi^ F4/80^+^ CD11b^Lo^	− Migrating Tissue Macro	**I.2** Ly6C^Hi^ F4/80^+^ CD11b^Lo^	− Migrating Tissue Macro
**II.1** Ly6C^Lo^ F4/80^+^ CD11b^Hi^	++ Mature BM-derived monocyte	**III** Ly6C^−^ F4/80^−^	− L-DC	**III** Ly6C^−^F4/80^−^	− Dendritic Cell	**II** Ly6C^Lo^ F4/80^+^	− & ++ Mature BM-derived Macro	**II** Ly6C^Lo^ F4/80^+/−^	− & + Dendritic Cell & Mature BM-derived Macro	**II** Ly6C^Lo^ F4/80^+^	− Dendritic Cell
**II.2** Ly6C^Lo^ F4/80^+^ CD11b^Lo^	− Patrolling monocyte	**IV** Ly6C^−^ F4/80^+^	− MZ or MMM Macro	**IV** Ly6C^−^F4/80^+/−^	− Dendritic Cell	**III** Ly6C^−^ F4/80^−^	− Dendritic Cell	**III** Ly6C^−^ F4/80^−^	− Dendritic Cell	**III** Ly6C^−^ F4/80^+/−^	− Dendritic Cell
**III** Ly6C^−^ F4/80^−^	− Patrolling monocyte			**V** Ly6C^−^ F4/80^Hi^	+ Red Pulp Macro	**IV** Ly6C^−^ F4/80^+^	− & + Dendritic Cell & Mature BM-derived Macro	**IV** Ly6C^−^ F4/80^Lo^	+ Mature BM-derived Macro		
**IV** Ly6C^−^ F4/80^+^	− & ++ Migrating Tissue Macro							**V** Ly6C^−^ F4/80^Hi^	++++ Lamina Propria Macro		

^1^ Based on data from Figure 2. ^2^ Based on data from Figure 3. ^3^ Based on data from Figure 5. ^4^ Based on data from Figure 6. ^5^ Based on data from Figure 7. ^6^ Relative MPO expression with blood neutrophil MPO expression set at ++++ (>6000 MFI), +++ (3000–4000 MFI), ++ (2000–3000 MFI), + (1500–2000 MFI), and − (<1500 MFI). ^7^ Macrophage abbreviation.

## Data Availability

Not applicable.

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
