# Peer review of "Myeloperoxidase as a Marker to Differentiate Mouse Monocyte/Macrophage Subsets"

_ijms, 2022, doi:10.3390/ijms23158246_

Round 1

Reviewer 1 Report

The goal of this study was to evaluate whether myeloperoxidase (MPO) could be used as a marker of pro-inflammatory monocyte/macrophage subsets, since specialized monocyte/macrophage subsets are difficult to differentiate based on cell surface markers. Overall, this is a well conducted study and the novelty of this paper relies in the description of the new, promising marker, that can be used as an additional marker for monocyte/macrophage subsets differentiation.I found only few minor deficiencies that need to be addressed. My concerns are outlined below:

1. Authors describe multiple populations of monocyte/macrophages in different organs, and in my opinion it would be valuable to add a table, that summarizes this novel data. This would make the paper more interesting and easier to process.

2. In the discussion section Authors state, that “There are no markers that can differentiate between embryonic and adult monocytes/macrophages nor are there markers that can differentiate monocytes from macrophages.” Since this is true for monocyte and macrophage, there are marker sets that enable the differentiation between different populations of resident macrophages of embryonic and bone marrow origin, at least for some organs (for example heart, as described by Epelmann, 10.1016/j.immuni.2013.11.019). Undoubtedly, new markers are needed, but the statement that the macrophage populations cannot be distinguished is in my opinion too much of a simplification.

3. I’ve found a couple of typos in the text – e.g. page 10, line 340 (endothelum instead of endothelium) or page 13, line 491 (granles instead of granules). Please check spelling carefully.

Author Response

Rebuttal

1) A table has been added to summarize MPO expression in various monocyte/macrophage subsets.

2) Thank you for the clarification. The statement referred to regarding distinguishing embryonic and BM-derived monocyte/macrophage subsets was deleted.

3) The typos have been corrected.

Reviewer 2 Report

Dear Editor, in the submitted work authors are using a combination of pan” monocyte/macrophage markers and flow cytometry, in order to evaluate whether myeloperoxidase (MPO) could be used as a marker of pro-inflammatory monocyte/macrophage subsets. The paper is well organized and provides new data, which may be of interest for a lot of scientists. For this reason I propose to accept it for publication. As minor importance comment I propose to authors to add some conclusions after discussion part.

Author Response

We have added a short conclusion as requested.